# Gain of channel function and modified gating properties in TRPM3 mutants causing intellectual disability and epilepsy

Evelien Van Hoeymissen[1,2†], Katharina Held[1,2†], Ana Cristina Nogueira Freitas[2†], Annelies Janssens[2], Thomas Voets[2‡*], Joris Vriens[1‡*]

[1]Laboratory of Endometrium, Endometriosis and Reproductive Medicine, Department of Development and Regeneration, Leuven, Belgium; [2]Laboratory of Ion Channel Research, VIB-KU Leuven Center for Brain and Disease Research, Belgium and Department of Molecular Medicine, Leuven, Belgium

**Abstract** Developmental and epileptic encephalopathies (DEE) are a heterogeneous group of disorders characterized by epilepsy with comorbid intellectual disability. Recently, two de novo heterozygous mutations in the gene encoding TRPM3, a calcium permeable ion channel, were identified as the cause of DEE in eight probands, but the functional consequences of the mutations remained elusive. Here we demonstrate that both mutations (V990M and P1090Q) have distinct effects on TRPM3 gating, including increased basal activity, higher sensitivity to stimulation by the endogenous neurosteroid pregnenolone sulfate (PS) and heat, and altered response to ligand modulation. Most strikingly, the V990M mutation affected the gating of the non-canonical pore of TRPM3, resulting in large inward cation currents via the voltage sensor domain in response to PS stimulation. Taken together, these data indicate that the two DEE mutations in TRPM3 result in a profound gain of channel function, which may lie at the basis of epileptic activity and neurodevelopmental symptoms in the patients.

**\*For correspondence:**
thomas.voets@kuleuven.vib.be (TV);
Joris.Vriens@kuleuven.be (JV)

[†]These authors contributed equally to this work
[‡]These authors also contributed equally to this work

**Competing interests:** The authors declare that no competing interests exist.

## Introduction

Transient Receptor Potential (TRP) channel TRPM3 is a calcium-permeable cation channel that can be activated by heat (*Vriens et al., 2011*) and by a variety of chemical ligands, including the endogenous neurosteroid pregnenolone sulfate (PS) (*Wagner et al., 2008*). TRPM3 is expressed in a large subset of mouse and human somatosensory neurons, where it is involved in the detection of noxious heat and the development of inflammatory pain (*Vriens et al., 2011*; *Vangeel et al., 2020*). Moreover, TRPM3 is expressed in several brain areas, including the choroid plexus, cerebellum, cortex and the hippocampal formation (*Grimm et al., 2003*; *Oberwinkler et al., 2005*; *Zamudio-Bulcock et al., 2011*), but its functional role in these areas is unknown. Recently, two de novo substitutions in TRPM3 (V990M and P1090Q) were identified as the cause of intellectual disability and epilepsy in eight probands with developmental and epileptic encephalopathy (DEE) (*Dyment et al., 2019*). Interestingly, clinical findings in these eight individuals also included one individual with an altered pain sensitivity and a second individual showing a modified heat sensitivity, even if pain sensitivity was not systematically tested (*Dyment et al., 2019*). However, the consequences of these mutations on TRPM3 function remained elusive. We here demonstrate that both mutations lead to significant gain-of-channel function, including increased basal activity and higher sensitivity to PS and heat. V990M exhibits further pronounced functional alterations, including anomalous activation of the alternative current through the voltage-sensor domain, reduced sensitivity to receptor-mediated inhibition and calcium-dependent inactivation, and lower sensitivity to block by the anticonvulsant primidone.

## Results and discussion

When performing Fura-2-based calcium imaging on transiently transfected HEK293T cells, we observed significantly higher intracellular $Ca^{2+}$ concentrations ($[Ca^{2+}]_i$) in cells expressing the two DEE mutants compared to wild type human TRPM3 (WT; GenBank: AJ505026.1), an effect that was much more pronounced in the V990M mutant (*Figure 1A,B*). Application of primidone or isosakuranetin, both potent TRPM3 antagonists (*Krügel et al., 2017*; *Straub et al., 2013*), reduced $[Ca^{2+}]_i$ in cells expressing WT or mutant TRPM3. In absolute terms, the antagonist-induced reduction in $[Ca^{2+}]_i$ was the largest for the V990M mutant, yet $[Ca^{2+}]_i$ did not fully return to the level of cells expressing WT (*Figure 1A,B* and *Figure 1—figure supplement 1*). Importantly, both mutants were expressed at similar levels as WT, as assessed based on the fluorescence signal of the C-terminally attached YFP (*Figure 1—figure supplement 1*). These results suggest that the DEE mutations lead to increased basal channel activity. In line herewith, whole-cell currents in response to voltage steps revealed increased current densities in cells expressing the DEE mutants compared to WT (*Figure 1—figure supplement 2*).

Next, we compared the responses of WT and DEE mutants to stimulation with agonist PS and with clotrimazole (Clt), an antifungal drug and known TRPM3 modulator (*Vriens et al., 2014*). In line with earlier studies (*Wagner et al., 2008*), we found that PS (40 µM) reversibly activated outwardly rectifying whole-cell currents in cells expressing WT, whereas application of Clt (10 µM) did not activate currents by itself but potentiated responses to PS (*Figure 1C,D* and *Figure 1—figure supplement 2*). In particular, application of PS in the presence of Clt provoked activation of a large inwardly rectifying current component, which has been attributed to activation of an alternative ion permeation pathway located in the voltage-sensing domain, distinct from the central pore (*Vriens et al., 2014*). The response pattern was strikingly altered in cells expressing the DEE mutants. In the case of V990M, whole-cell current densities in response to PS were significantly larger compared to WT, and notably, exhibited a prominent inwardly rectifying current component. In addition, application of Clt induced robust currents in the absence of PS, whereas currents in the combined presence of Clt and PS were similar in amplitude and shape as WT (*Figure 1E,F* and *Figure 1—figure supplement 2*). In the case of P1090Q, whole-cell current densities in response to PS were also significantly larger compared to WT, but lacked the inwardly rectifying component observed in the V990M mutant. Clt did not activate currents in cells expressing P1090Q, and in contrast to WT, inhibited the response to PS (*Figure 1G,H* and *Figure 1—figure supplement 2*). Taken together, these results indicate that both DEE mutants lead to significant changes in channel gating, including increased basal activity and pronounced alterations in ligand responses.

To further assess the enhanced response to PS stimulation of the DEE mutants, we compared their apparent affinity to PS by measuring $[Ca^{2+}]_i$ responses to stepwise increases in PS concentrations. We found that the concentration-response curve for both mutants was shifted to significantly lower concentrations compared to WT. Notably, whereas a PS concentration of 10 µM was required to induce a detectable response in cells expressing WT TRPM3, we observed robust calcium responses at concentrations as low as 100 nM for V990M and 1 µM for P1090Q. Moreover, the maximal increase in $[Ca^{2+}]_i$ to saturating PS concentrations was significantly higher in P1090Q expressing cells compared to WT (*Figure 2A,B*). Thus, both DEE mutants show increased responses to the neurosteroid PS.

TRPM3 is a temperature-sensitive channel, activated upon heating (*Vriens et al., 2011*). To address whether the DEE mutations affect the channel's response to heat, we compared changes in $[Ca^{2+}]_i$ in cells expressing WT or mutant TRPM3 upon exposure to a heat ramp from 23°C to 40°C. Compared to non-transfected cells, we measured a significantly larger increase in $[Ca^{2+}]_i$ in cells expressing WT or mutant TRPM3. Notably, the amplitude of the heat-induced response was significantly larger in cells expressing the DEE mutants compared to WT (*Figure 2C,D*).

TRPM3 activity is inhibited upon activation of G protein-coupled receptors, via a mechanism that involves direct binding of the $G_{\beta\gamma}$ of trimeric G-proteins to the channel (*Badheka et al., 2017*; *Dembla et al., 2017*; *Quallo et al., 2017*). To evaluate whether $G_{\beta\gamma}$-dependent modulation is altered in the DEE mutations, we co-transfected HEK293T cells with the µ-opioid receptor and WT or mutant TRPM3, and evaluated the effect of the selective agonist DAMGO on PS-activated whole-cell currents. In cells expressing WT or P1090Q, application of 1 µM DAMGO induced a complete and rapidly reversible inhibition of inward and outward currents, whereas V990M was only partly

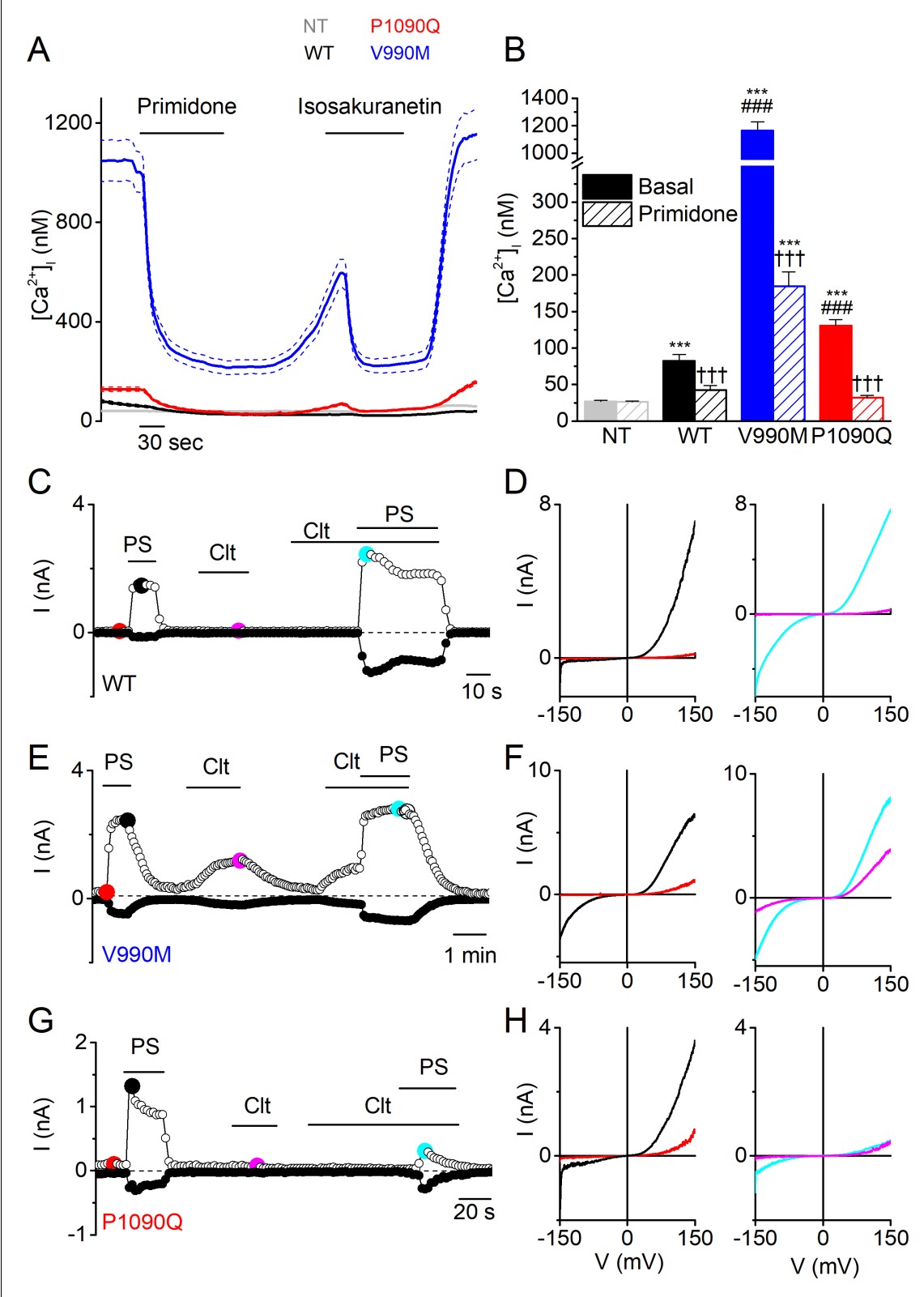

**Figure 1.** Elevated basal activity in HEK293T cells expressing TRPM3 DEE mutants. (**A**) Time course of intracellular calcium concentrations ([Ca$^{2+}$]$_i$) (± SEM) upon application of the TRPM3 inhibitors primidone (100 µM) and isosakuranetin (50 µM) for WT (n = 230), P1090Q (n = 163) and V990M (n = 79) transfected HEK293T cells, and non-transfected (NT) cells (n = 93) (N = 3 independent experiments). (**B**) Basal intracellular calcium concentrations in the absence (full bars) and presence of primidone (open bars). Data are represented as mean ± SEM, using a Kruskal-Wallis ANOVA

*Figure 1 continued on next page*

*Figure 1 continued*

with Dunn's posthoc test, where ***=versus NT; ###=versus basal WT; †††=versus basal of the same mutant/WT. For WT: ***p=$7.5\times10^{-5}$, ††† p=$9.4\times10^{-6}$; for V990M: ***p=$1.6\times10^{-60}$, ###: p=$1.4\times10^{-42}$, ††† p=$7.3\times10^{-11}$; for P1090Q: ***p=$1.5\times10^{-19}$, ###: p=$4.9\times10^{-7}$, ††† p=$1.6\times10^{-19}$.(C, E and G) Amplitude of currents at a holding potential of +80 mV and –80 mV (measured with voltage ramps) upon application of PS (40 µM), Clt (10 µM) and co-application of PS and Clt for WT (n = 10) (C), V990M (n = 7) (E) and P1090Q (n = 9) (G). (D, F and H) Current-voltage relationships at the time points indicated in (C), (E) and (G).

The online version of this article includes the following figure supplement(s) for figure 1:

**Figure supplement 1.** Increased basal intracellular calcium concentrations in DEE mutants that are not linked to increased expression levels of DEE mutants.

**Figure supplement 2.** Biophysical characterization of the V990M and P1090Q substitution in hTRPM3 indicate the substitutions as a gain of function mutation.

inhibited (*Figure 2E–I*). The DAMGO concentration for half-maximal inhibition of the PS-activated currents shifted from 4.0 ± 0.6 nM for WT to 40 ± 10 nM for V990M (*Figure 2—figure supplement 1*). Note that DAMGO was without effect on TRPM3 currents in cells that were not co-transfected with the µ-opioid receptor (*Figure 2—figure supplement 1*). A difference in sensitivity was also found for the anticonvulsant drug primidone, which at a concentration of 25 µM caused a full inhibition of PS-activated inward and outward currents mediated by WT or P1090Q, but blocked currents mediated by V990M by only ~50% (*Figure 2E–G,I*). A similar difference in primidone sensitivity was observed using Fura-2-based calcium imaging (*Figure 2—figure supplement 1*).

When switching from the standard, $Ca^{2+}$-free extracellular solution to a solution containing 1 mM $Ca^{2+}$, PS-activated currents mediated by WT or by P1090Q undergo time-dependent desensitization (*Vriens et al., 2014*; *Held et al., 2015*). In contrast, PS-activated currents mediated by V990M remained stable in the presence of extracellular $Ca^{2+}$, indicating reduced sensitivity to $Ca^{2+}$-dependent desensitization in this mutant (*Figure 2J–M*).

The PS-induced whole-cell currents mediated by V990M showed a prominent inwardly rectifying current component. A similar inwardly rectifying current component can also be activated in WT TRPM3 when PS is applied in the presence of Clt. In earlier work, we have demonstrated that this inwardly rectifying current component represents ion flux through an alternative ion permeation pathway located in the voltage sensor domain of TRPM3, which can be distinguished from the central pore based on its voltage dependence, insensitivity to pore block by $La^{3+}$ and ion selectivity (including a lower permeability for monomethylammonium ($MMA^+$) compared to $Na^+$) (*Vriens et al., 2014*; *Held et al., 2015*; *Held et al., 2018*). Notably, the V990M mutation is located in close vicinity of Asp988 and Gly991, which we recently identified as critical determinants of the alternative ion permeation pathway (*Figure 3—figure supplement 1*; *Held et al., 2018*). We therefore hypothesized that gating of the alternative ion permeation pathway is facilitated in V990M, such that it can be activated by PS even when Clt is not co-applied. We found that the PS-activated current in the V990M mutant showed a bimodal voltage dependence (*Figure 3B*), its inward current component was resistant to block by the central pore blocker $La^{3+}$ (*Figure 3F,G*), and inward currents were reduced when extracellular $Na^+$ was replaced by $MMA^+$ (*Figure 3H*). Taken together, these data indicate that the V990M mutation leads to a gain of function at the level of the alternative ion permeation pathway.

Considering that all reported DEE patients were heterozygous for the TRPM3 substitutions (*Dyment et al., 2019*), and that TRPM3 is functional as a tetramer, it can be expected that patient cells express a mixture of WT and mutant channel subunits, potentially leading to the formation of heteromultimeric channels with variable stoichiometry. To mimic the heterozygous condition in vitro, we performed a limited number of experiments in cells co-transfected with a mixture of cDNA encoding WT and mutant TRPM3 in a 1:1 ratio. The current densities of PS-induced inward currents in cells expressing a WT:V990M mixture was intermediate between cells expressing only WT or only V990M (*Figure 4A,B*). Moreover, the PS-activated currents exhibited the typical inwardly rectifying current component (*Figure 4C,D*). Finally, in contrast to WT but like V990M, Clt (10 µM) activated robust currents in cells expressing a WT:V990M mixture (*Figure 4A,B*). Next, the WT:P1090Q co-transfected cells showed PS-induced current densities that were intermediate between WT and P1090Q transfected cells (*Figure 4E,F*) and showed a shift in the rectification pattern of the PS-induced currents that was different from the homozygote situation (*Figure 4G*). Moreover, the effect

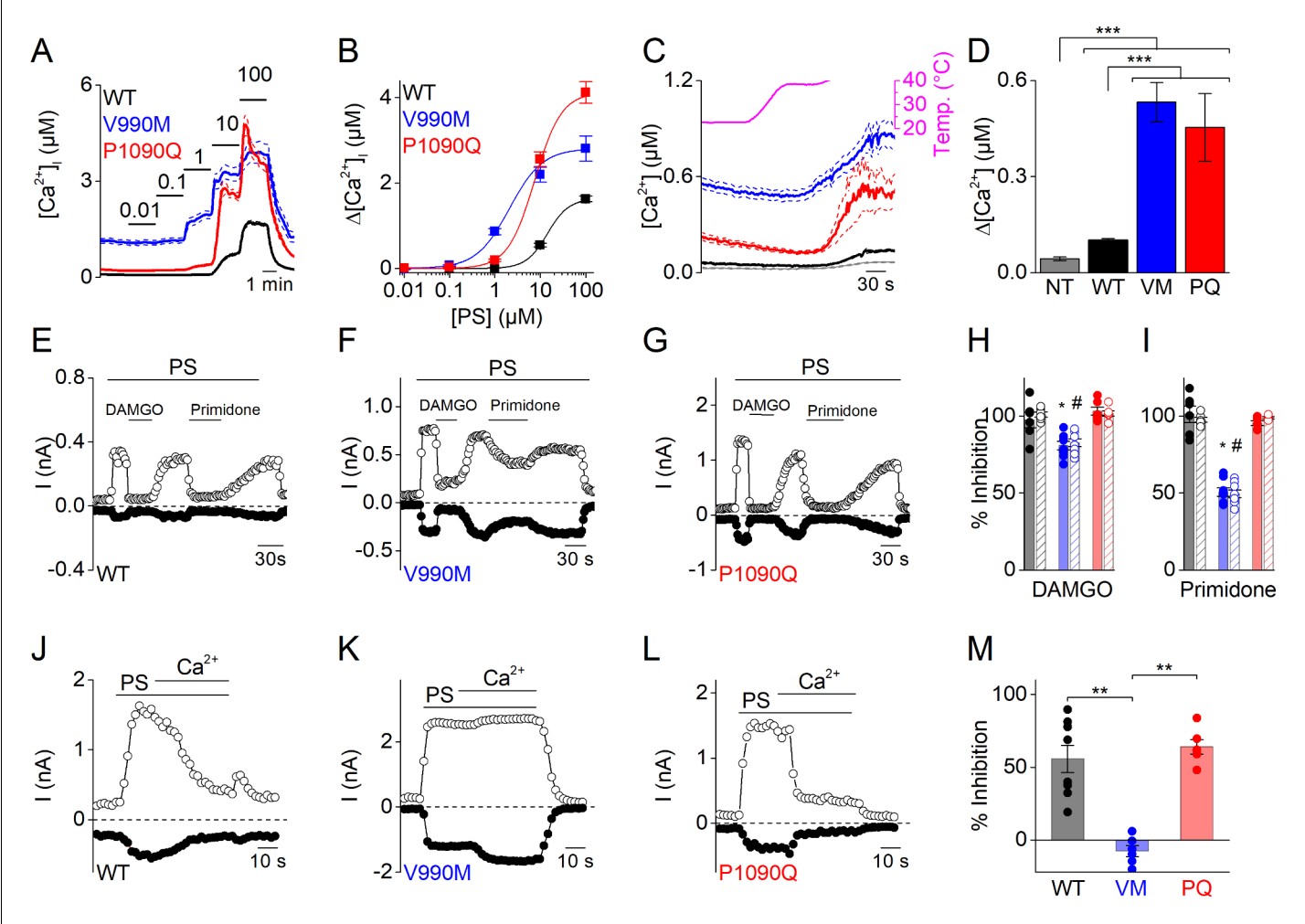

**Figure 2.** Altered sensitivity of DEE mutants for thermal stimulation and pharmacological modulation. (**A**) Time course of $[Ca^{2+}]_i$ (± SEM) upon application of the TRPM3 agonist PS in stepwise increasing dose (0.01–100 µM) for WT (n = 615), V990M (n = 130) and P1090Q (n = 196) (N = 3 independent experiments). (**B**) PS concentration-response curves for WT ($EC_{50}$ = 14.3 ± 5.8 µM), V990M ($EC_{50}$ = 2.1 ± 0.4 µM), and P1090Q ($EC_{50}$ = 7.5 ± 1.4 µM). (**C**) Time course of $[Ca^{2+}]_i$ for NT (gray, n = 46), WT (black, n = 148), V990M (blue, n = 259) and P1090Q (red, n = 271) when applying a heat ramp (magenta). Analysis of 3 independent experiments, where the data are represented as mean ± SEM. (**D**) Corresponding amplitudes of the temperature response, represented as mean ± SEM, using a Kruskal-Wallis ANOVA with Dunn's posthoc test (***). WT, V990M and P1090Q transfected cells had a significant larger amplitude compared to NT cells (p=$4.6\times10^{-5}$, p=$2.5\times10^{-22}$ and p=$1.2\times10^{-22}$, respectively). V990M and P1090Q transfected cells had a significant larger amplitude compared to WT cells (p=$8.3\times10^{-14}$ and p=$2.8\times10^{-14}$). The amplitudes of P1090Q and V990M were not significantly different (p=1). (**E–G**) Amplitude of currents at +80 mV and −80 mV (measured during voltage ramps) upon application of PS (40 µM) with co-application of the µ-opioid receptor agonist DAMGO (1 µM) or the TRPM3 inhibitor primidone (25 µM) for WT (n = 6) (**E**), V990M (n = 8) (**F**) and P1090Q (n = 6) (**G**), in cells co-expressing the µ-opioid receptor. (**H–I**) Percentage inhibition of PS-induced currents upon application of DAMGO (+80 mV: WT versus V990M (p=$3.4\times10^{-5}$), WT versus P1090Q (p=0.98) and V990M versus P1090Q (p=$2.3\times10^{-5}$); −80 mV: WT versus V990M (p=0.01), WT versus P1090Q (p=0.54) and V990M versus P1090Q (p=$8.7\times10^{-4}$). (**H**) And primidone (+80 mV: WT versus V990M (p=0), WT versus P1090Q (p=0.88) and V990M versus P1090Q (p=0); −80 mV: WT versus V990M (p=0), WT versus P1090Q (p=0.52) and V990M versus P1090Q (p=$6.1\times10^{-8}$). (**I**) For WT (black), V990M (blue) and P1090Q (red). The filled and shaded bars represent the current inhibition at −80 mV and +80 mV, respectively. A Kruskal-Wallis ANOVA with Dunn's posthoc test was used. Data are represented as mean ± SEM and scatter plots for each individual cell. (**J–L**) Amplitude of currents at +80 mV and −80 mV (measured during voltage ramps) upon application of PS (40 µM) in the presence of 1 mM extracellular calcium for WT (n = 8) (**J**), V990M (n = 6) (**K**) and P1090Q (n = 6) (**L**). (**M**) Percentage inhibition upon calcium application for WT, V990M (VM) and P1090Q (PQ) (mean ± SEM and scatter plots for each individual cell). **: A Kruskal-Wallis ANOVA with Dunn's posthoc test was used, where the amplitude of V990M transfected cells compared to WT and P1090Q transfected cells were significantly different (p=0.009 and p=0.005, respectively). The amplitudes of P1090Q and WT transfected cells were not significantly different (p=1).

The online version of this article includes the following figure supplement(s) for figure 2:

**Figure supplement 1.** Characterization of DAMGO and primidone block on PS-induced currents.

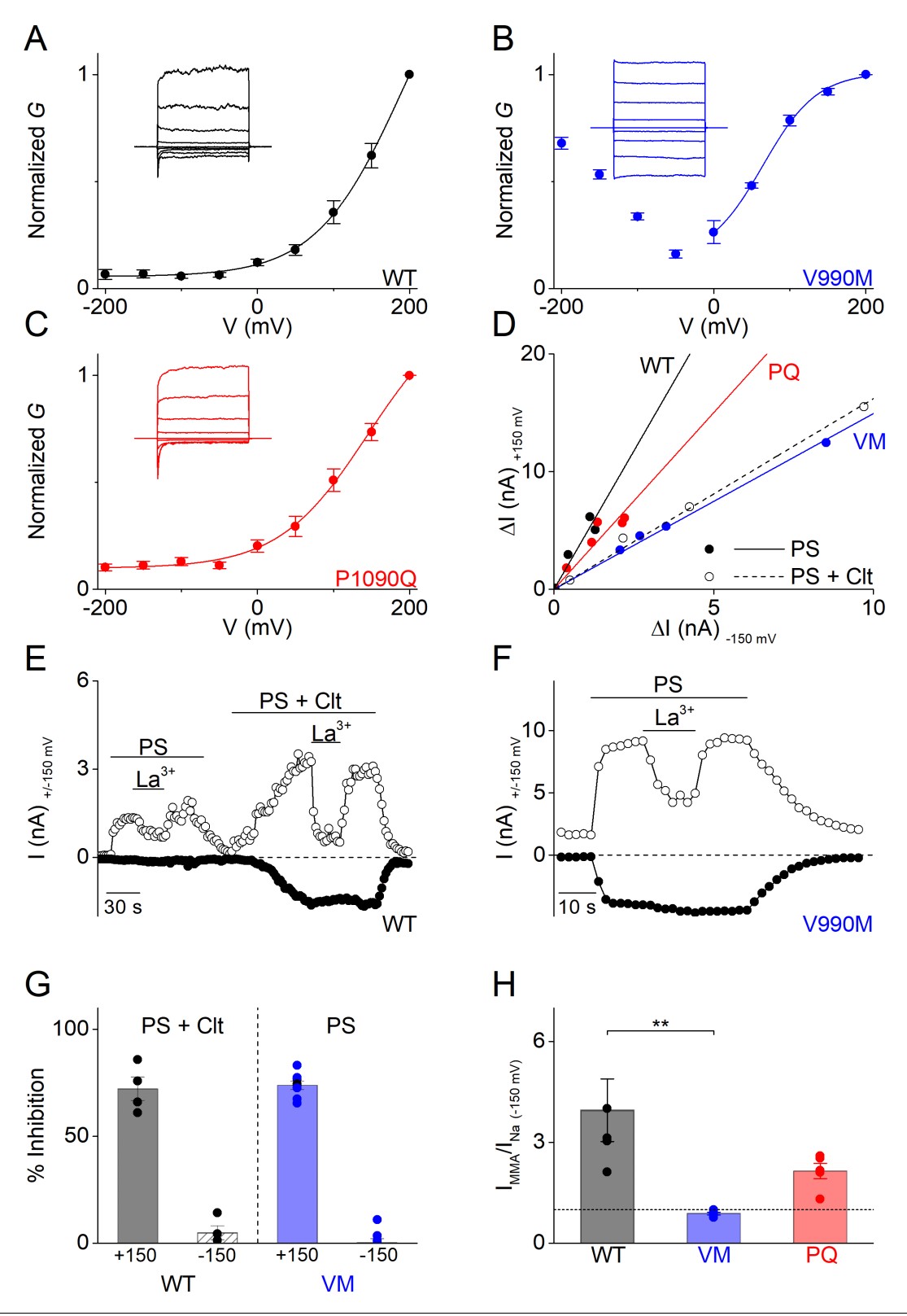

**Figure 3.** Altered gating of the alternative pore in V990M (A–C) *G-V* plots of PS-activated currents for (A) WT (black), (B) V990M (blue) and (C) P1090Q (red). Currents measured during voltage-steps ranging from −200 mV to +200 mV, separated by steps of +50 mV. Representative currents are shown as insets in each graph; n = 6 for each experiment. (D) Rectification pattern of PS (40 µM) (full circle and line) and PS + Clt (10 µM)-induced (open circle and dashed line) currents for WT, V990M and P1090Q. Data points are derived by plotting the current increase at +150 mV *versus* the current increases

*Figure 3 continued on next page*

*Figure 3 continued*

at −150 mV; n ≥ 4 for each dataset. (**E**) Time course of WT TRPM3 whole-cell currents at ± 150 mV upon application of PS (40 µM) and Lanthanum (La$^{3+}$; 10 µM) or PS + Clt and La$^{3+}$. (**F**) Time course of V990M mutant whole-cell currents at ± 150 mV upon application of PS and La$^{3+}$. (**G**) Relative La$^{3+}$ block calculated from experiments as in E) and F) for WT (black) in presence of PS + Clt (n = 4) and for V990M (blue, n = 8) in presence of PS alone (mean ± SEM and scatter plot for each individual cell). (**H**) Relative PS-induced currents at −150 mV carried by monomethylammonium (MMA$^+$) in WT (black), V990M (blue) and P1090Q (red). MMA$^+$ currents were normalized to the currents carried by Na$^+$; PS (40 µM) and n = 5 for all experiments (mean ± SEM and scatter plot for each individual cell). ** One-way ANOVA with Tukey's posthoc test (WT versus V990M: p=0.005; WT versus P1090Q: p=0.09 and V990M versus P1090Q: p=0.28).

The online version of this article includes the following figure supplement(s) for figure 3:

**Figure supplement 1.** Homology model illustrating the different positions of the DEE mutations Homology model of TRPM3 based on the published cryo-EM structure of TRPM4 (pdb code: 6bcj).

of Clt pre-incubation was different in co-transfected cells. WT:P1090Q cells showed first a potentiation of the PS responses by pre-incubation of Clt, followed by a time-dependent current inhibition. This was in contrast to the block of PS-induced currents by pre-incubation with Clt for the P1090Q mutant in isolation (*Figure 4H*). Taken together, these findings indicate that also in the heterozygous condition both DEE mutations lead to a gain of channel function. However, since at this point we do not know whether the mutations affect the formation of channel tetramers, further experiments are required to determine the precise stoichiometry of WT and mutant subunits in the TRPM3 channels of DEE patients.

In conclusion, our results indicate that two human mutations in the TRPM3 gene associated with DEE give rise to channels with substantially altered functional properties. Whereas the V990M and P1090Q mutations have various differential effects on several aspects of TRPM3 gating, both can be considered as strong gain-of-function mutants, with increased inward cation currents and Ca$^{2+}$ influx under basal condition or when stimulated with heat or the endogenous neurosteroid PS. We hypothesize that the increased calcium influx and depolarizing channel activity may lie at the basis of seizure development and neurodevelopmental symptoms in DEE patients.

# Materials and methods

## Key resources table

| Reagent type (species) or resource | Designation | Source or reference | Identifiers | Additional information |
|---|---|---|---|---|
| Cell line (human) | HEK293T | Dr. S Roper, University of Miami school of medicine Depart. of physiology and biophysics, 4044 Miami FL 33136 | ATCCCRL-3216 | |
| Recombinant DNA reagent | hOPMR1 (plasmid) | Received from Missouri university | Catalogue # OPM10FN00 | In pCDNA3 plasmid. |
| Recombinant DNA reagent | TRPM3 (plasmid) | Received from C Harteneck - Berlin | GenBank: AJ505026.1 | In pCDNA3/V5/ his/plasmid tagged with YFP. |
| Recombinant DNA reagent | TRPM3 with V990M mutation (plasmid) | This paper | GenBank: AJ505026.1 modified by V990M mutation | In pCDNA3/V5/ his/plasmid tagged with YFP |
| Recombinant DNA reagent | TRPM3 with P1090Q mutation (plasmid) | This paper | GenBank: AJ505026.1 modified by P1090Q mutation | In pCDNA3/V5/his/plasmid tagged with YFP |
| Commercial assay or kit | TransIT-293 Transfection Reagent | Mirus | Catalogue # MIR 2700 | |

*Continued on next page*

*Continued*

| Reagent type (species) or resource | Designation | Source or reference | Identifiers | Additional information |
|---|---|---|---|---|
| Chemical compound, drug | Pregnenolone Sulfate | Sigma-Aldrich | Catalogue # P162 | TRPM3 agonist |
| Chemical compound, drug | DAMGO | Sigma-Aldrich | Catalogue # E7384 | μ-opioid receptor agonist |
| Chemical compound, drug | Isosakuranetin | Carl Roth | Catalogue # 7498.1 | TRPM3 inhibitor |
| Chemical compound, drug | Clotrimazole | Sigma-Aldrich | Catalogue # C6019 | TRPM3 modulator |
| Chemical compound, drug | Primidone | Sigma-Aldrich | Catalogue # P7295 | TRPM3 inhibitor |
| Chemical compound, drug | Fura-2-acetoxymethyl ester | Alexis Biochemicals | Catalogue # ENZ-52006 | Calcium indicator |
| Software, algorithm | OriginPro 8.6 | OriginLab Corporation, USA | RRID:SCR_014212 | Data analysis and statistical analysis |
| Software, algorithm | IgorPro 6.2 | WaveMetrics, USA | RRID:SCR_000325 | Data analysis |
| Software, algorithm | ImageJ | https://imagej.net/ | RRID:SCR_003070 | Data analysis |
| Software, algorithm | NIS-Elements | Nikon | RRID:SCR_014329 | Acquisition $Ca^{2+}$-imaging data |
| Software, algorithm | PatchMaster Pro | HEKA Elektronik, Lambrecht, Germany | | Acquisition patch clamp data |

## Cell culture

HEK293T cells (*Graham et al., 1977*) were cultured as described previously (*Vriens et al., 2007*) and used up in passage number 14. HEK293T cells were tested for the lack of mycoplasma and transiently transfected with 2 μg of DNA using TransIT transfection reagent (Mirus) 36–48 hr before the measurements. In case of co-cultures a ratio of 1:1 between hTRPM3-YFP WT and mutants-YFP DNA was used.

## Site-directed mutagenesis

All mutants were obtained by the standard PCR overlap extension method using hTRPM3 directly linked to YFP from pCAGGS/IRES-GFP vector (*Vriens et al., 2007*). Accuracy of all mutant sequences was verified by sequencing of the entire DNA constructs.

## Fluorescence imaging

Changes in intracellular calcium concentration were monitored using ratiometric Fura-2-based fluorimetry. Cells were loaded with 2 μM Fura-2-acetoxymethyl ester (Alexis Biochemicals) for 30 min at 37°C. Fluorescence was measured during alternating illumination at 340 and 380 nm using Eclipse Ti (Nikon) fluorescence microscopy system, and absolute calcium concentration was calculated from the ratio of the fluorescence signals at these two wavelengths (R = $F_{340}/F_{380}$) as $[Ca^{2+}]$ =$K_m \times$ (R-$R_{min}$)/($R_{max}$-R), where $K_m$, $R_{min}$ and $R_{max}$ were estimated from in vitro calibration experiments with known calcium concentrations. The bath solution contained (in mM) 138 NaCl, 5.4 KCl, 2 $CaCl_2$, 2 $MgCl_2$, 10 glucose, and 10 HEPES, pH 7.4. Pregnenolone sulfate, clotrimazole and

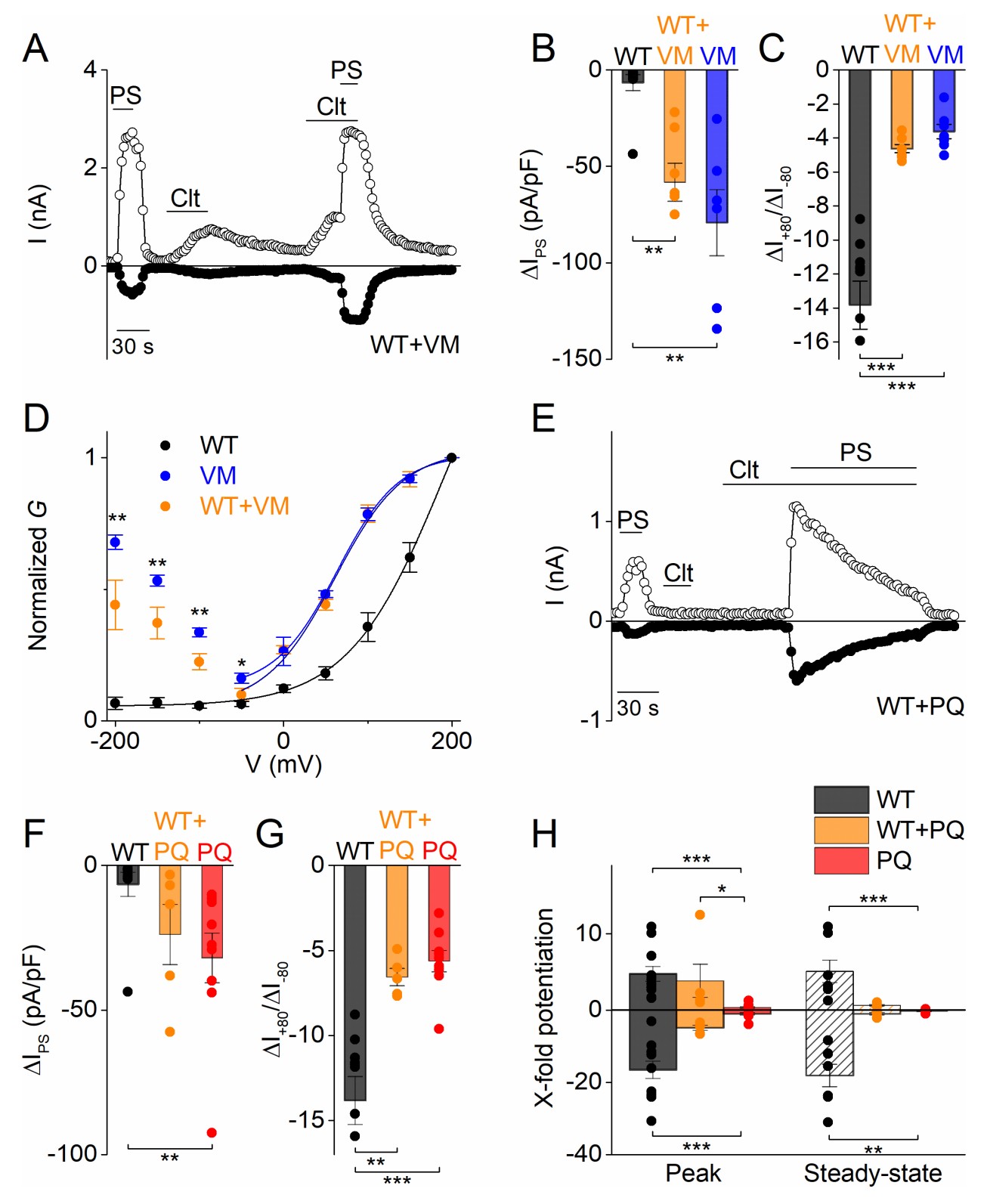

**Figure 4.** Heterozygous effects of DEE mutants. (**A**) Time course of whole-cell currents at ± 80 mV recorded in HEK293T cells transiently co-transfected with WT and V990M mutant DNA (1:1) upon application of PS (40 μM), Clt (10 μM) or PS + Clt. (**B**) Current densities at a holding potential of −80 mV (measured with voltage ramps) upon application of PS (40 μM) for WT (n = 10) (black), VM (n = 7) (blue) or WT + VM (1:1) (n = 7) (orange). The current densities for WT versus V990M (p=0.002) and for WT versus V990M + WT (p=0.008) were significantly different. The current densities for V990M versus

*Figure 4 continued on next page*

*Figure 4 continued*

V990M + WT were not significantly different (p=1). (C) Same as in (B) but for current amplitude ratios of +80 mV /-80 mV. The ratios for WT versus V990M (p=$2.3\times10^{-6}$) and for WT versus V990M + WT (p=$1\times10^{-5}$) were significantly different. The ratios for V990M versus V990M + WT were not significantly different (p=0.81) (D) *G-V* plots for PS-activated WT TRPM3 (black), V990M (blue) and WT + V990M (1:1) (orange). Data points were obtained with a step protocol ranging from −200 mV to +200 mV with +50 mV steps; n ≥ 6 for each experiment. (E) Time course of whole-cell currents at ±80 mV recorded in HEK293T cells transiently co-transfected with WT and P1090Q (1:1) upon application of PS (40 µM), Clt (10 µM) or PS + Clt. (F) Similar as in (B) but for WT (n = 10) (black), P1090Q (n = 9) (red) or co-transfected WT + P1090Q (1:1) (n = 5) (orange). The current densities for WT versus P1090Q (p=0.004) were significantly different. The current densities for WT versus P1090Q + WT (p=0.07) and P1090Q versus P1090Q + WT were not significantly different (p=0.55) (G) Similar as in (C) but for WT (n = 10) (black), P1090Q (n = 9) (red) or co-transfected WT + P1090Q (1:1) (n = 5) (orange). The ratios for WT versus P1090Q (p=$4.2\times10^{-5}$) and for WT versus P1090Q + WT (p=0.001) were significantly different. The ratios for P1090Q versus P1090Q + WT were not significantly different (p=0.86). (H) X-fold potentiation at peak (+80 mV: for WT versus P1090Q (p=$3.5\times10^{-5}$), WT versus P1090Q + WT (p=1) and P1090Q versus P1090Q + WT (p=0.031); −80 mV: for WT versus P1090Q (p=$1.4\times10^{-5}$), WT versus P1090Q + WT (p=0.37) and P1090Q versus P1090Q + WT (p=0.13)) and steady-state (+80 mV: for WT versus P1090Q (p=$6\times10^{-4}$), WT versus P1090Q + WT (p=0.13) and P1090Q versus P1090Q + WT (p=0.35); −80 mV: for WT versus P1090Q (p=0.002), WT versus P1090Q + WT (p=0.064) and P1090Q versus P1090Q + WT (p=0.86)) conditions of Clt-potentiated PS-currents in WT (black), P1090Q (red) or co-transfected with WT + P1090Q (1:1) (orange); n ≥ 4. All bar plots are represented as mean ± SEM and scatter plot for each individual cell. Kruskal-Wallis ANOVA with Dunn's posthoc test for panel B, F and H. One-way ANOVA with Tukey's posthoc test for panel C and G. Mann-Whitney test was used for comparison of V990M versus V990M + WT in panel D.

primidone were obtained from Sigma-Aldrich, isosakuranetin was obtained from Carl Roth. The chemical ligands were dissolved in bath solution from a stock diluted in DMSO.

## Whole cell patch clamp recordings

Standard whole-cell patch-clamp recordings were performed with an EPC-10 amplifier and the PatchMasterPro Software (HEKA Elektronik, Lambrecht, Germany). Current measurements were performed at a sampling rate of 20 kHz and currents were digitally filtered at 2.9 kHz. In all measurements, 70% of the series resistance was compensated. The standard internal solution contained (in mM): 100 CsAsp, 45 CsCl, 10 EGTA, 10 HEPES, 1 MgCl$_2$ (pH 7.2 with CsOH) or 140 potassium gluconate, 5 EGTA, 1 MgCl$_2$, 10 HEPES and 2 NaATP (pH 7.3 with CsOH) for the measurements of G$_{\beta\gamma}$ mediated inhibition and the standard extracellular solution contained (in mM): 150 NaCl, 1 MgCl$_2$, 10 HEPES (pH 7.4 with NaOH). The standard patch pipette resistance was between 2 MΩ and 4 MΩ when filled with pipette solution. In experiments allowing Ca$^{2+}$-dependent desensitization of TRPM3, Mg$^{2+}$ was replaced by Ca$^{2+}$ in the extracellular solution, and Cs$^+$ was replaced by Na$^+$ in the pipette solution. Pregnenolone sulfate, clotrimazole primidone and [D-Ala$^2$, N-Me-Phe$^4$, Gly$^5$-ol]-enkephalin acetate salt (DAMGO) were obtained from Sigma-Aldrich and isosakuranetin was obtained from Carl Roth. The chemical ligands were dissolved in bath solution from a stock diluted in DMSO.

## Statistics

Sample sizes were defined as minimum n = 5 for patch clamp experiments on HEK293T cells. One-way ANOVA power analysis was used for calculations of sample sizes. Parameters were taken as follows: statistical significance was tested for a 20% difference between groups with an α of 0.05 and a power of 0.9. For Calcium-fluorimetry measurements, no power analysis was performed, as each recording allowed to test more than 100 individual HEK293T cells.

Electrophysiological data were analyzed using IgorPro 6.2 (WaveMetrics, USA), WinASCD (Guy Droogmans, Leuven) and OriginPro 8.6 (OriginLab Corporation, USA). OriginPro 8.6 was further used for statistical analysis and data display. All data sets were tested for normality and the Student's paired, two-tailed *t*-test or the Mann-Whitney U test were used for statistical comparison between two different data sets. For comparison between multiple data sets One-way ANOVA with Tukey's posthoc test or Kruskal-Wallis ANOVA with Dunn's posthoc test were performed. P values below 0.05 were considered as significantly different. Data points represent means ± SEM of the given number (n) of identical experiments. No exclusion of statistical outliers was performed in this study.

Conductance-voltage (*G-V*) curves were fitted with a Boltzmann function of the form:

$$G(V) = \frac{G_{max}}{1 + \exp\left(zF\frac{(V_{1/2}-V)}{RT}\right)},$$

where $z$ is the apparent gating charge, $V_{1/2}$ the potential for half-maximal activation, $G_{max}$ the maximal conductance, $F$ the Faraday constant, $R$ the gas constant and $T$ the absolute temperature. Experiments were performed at room temperature.

## Acknowledgements

We thank all the members of the Laboratory of Ion Channel Research and the Laboratory of Endometrium, Endometriosis and Reproductive Medicine at the KU Leuven, for their helpful discussions and comments.

## Additional information

### Funding

| Funder | Grant reference number | Author |
|---|---|---|
| Fonds Wetenschappelijk Onderzoek | G.084515N | Joris Vriens |
| Fonds Wetenschappelijk Onderzoek | G.0B1819N | Joris Vriens |
| Fonds Wetenschappelijk Onderzoek | G.0565.07 | Thomas Voets Joris Vriens |
| Fonds Wetenschappelijk Onderzoek | G.0825.11 | Thomas Voets Joris Vriens |
| KU Leuven | C1-TRPLe | Thomas Voets |
| Fonds Wetenschappelijk Onderzoek | 12U7918N | Katharina Held |

The funders had no role in study design, data collection and interpretation, or the decision to submit the work for publication.

### Author contributions

Evelien Van Hoeymissen, Data curation, Visualization, Methodology, Writing - original draft, Writing - review and editing; Katharina Held, Data curation, Supervision, Validation, Visualization, Methodology, Writing - original draft, Writing - review and editing; Ana Cristina Nogueira Freitas, Data curation, Validation, Visualization, Writing - original draft, Writing - review and editing; Annelies Janssens, Data curation; Thomas Voets, Conceptualization, Formal analysis, Supervision, Funding acquisition, Investigation, Methodology, Writing - original draft, Writing - review and editing; Joris Vriens, Conceptualization, Resources, Supervision, Funding acquisition, Validation, Investigation, Visualization, Methodology, Writing - original draft, Project administration, Writing - review and editing

### Author ORCIDs

Evelien Van Hoeymissen (iD) https://orcid.org/0000-0003-3897-8998
Katharina Held (iD) https://orcid.org/0000-0002-1727-9517
Thomas Voets (iD) https://orcid.org/0000-0001-5526-5821
Joris Vriens (iD) https://orcid.org/0000-0002-2502-0409

### Decision letter and Author response

Decision letter https://doi.org/10.7554/eLife.57190.sa1
Author response https://doi.org/10.7554/eLife.57190.sa2

## Additional files

### Supplementary files
• Transparent reporting form

### Data availability
All data generated or analysed during this study are included in the manuscript and supporting files.

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
