## [Decision Letter]

**Acceptance summary:**

The physiological function of TRPM3 ion channels remains poorly understood. In this manuscript, Evelien Van Hoeymissen and colleagues functionally characterize recently reported single-point mutants of the TRPM3 channel and demonstrate that they produce gain-of-function phenotypes. Since these mutants are involved in producing epileptogenic activity, the results point towards a physiological explanation of the pathology and indicate an involvement of TRPM3 in neuronal excitability.

**Decision letter after peer review:**

Thank you for submitting your article "Gain of channel function and modified gating properties in TRPM3 mutants causing intellectual disability and epilepsy" for consideration by *eLife*. Your article has been reviewed by three peer reviewers, including Leon D Islas as the Reviewing Editor and Reviewer #1, and the evaluation has been overseen by Kenton Swartz as the Senior Editor The following individual involved in review of your submission has agreed to reveal their identity: Haoxing Xu (Reviewer #3).

The reviewers have discussed the reviews with one another and the Reviewing Editor has drafted this decision to help you prepare a revised submission.

Summary:

This manuscript provides a functional characterization of recently described mutations in TRPM3 channels that were proposed to explain a pathological neuronal phenotype. The experiments shown here demonstrate that these mutations produce a gain of function, which the authors suggest could be responsible for the increased excitability associated with the described pathology. These findings are of general relevance since they demonstrate, if indirectly, that TRPM3 channels serve important functions in the central nervous system. The characterization of the mutations' effects is thorough and the experiments are well conducted.

Essential revisions:

1) Overall, it is convincing that both V990M and P1090Q mutations confer some gain-of-function channel properties. However, it is well known that in the heterologous overexpression systems, the expression levels of the wild-type (WT) and mutant TRP channels may sometimes affect their ligand sensitivities and basal activities. What is the evidence that the expression levels of the V990M and P1090Q mutants are comparable to that of the WT channel? When comparing current magnitude between WT and mutants (Figure 1—figure supplement 1), data should be presented as averages of current density, not absolute current, since expression levels surely vary between cells and between mutants. Are the current densities of the maximal currents (i.e., PS+clt-induced currents) comparable for WT and mutant channels? Whereas representative traces were shown, the average current densities under various agonist conditions should be provided.

2) In Figure 1C, it would be more informative to present the fraction of calcium signal inhibited by primidone instead of -delta Ca.

3) Regarding the WT:mutant mixing experiments, it’s not clear how these data can be interpreted. The channel composition in humans is unknown; thus assuming that a heterozygous condition implies heteromeric channels is unsupported. The data presented in Figure 4 only indicate that the effects of the mutations can be conferred by a limited number of subunits, if the mutant can form tetramers with WT subunits (in 1:1 ratio), which is also not demonstrated and cannot be assumed a priori. If these data remain in the paper, these possibilities and limitations of the interpretation put forward should be clearly stated and discussed.

---

## [Author Response]

Essential revisions:1) Overall, it is convincing that both V990M and P1090Q mutations confer some gain-of-function channel properties. However, it is well known that in the heterologous overexpression systems, the expression levels of the wild-type (WT) and mutant TRP channels may sometimes affect their ligand sensitivities and basal activities. What is the evidence that the expression levels of the V990M and P1090Q mutants are comparable to that of the WT channel?

The potential difference in expression level is indeed a strong concern that is raised by the reviewers and could be an alternative plausible explanation for the elevated intracellular calcium concentrations in the two DEE mutants. Therefore, we have performed additional analysis of the microfluorimetric experiments and have checked the expression levels of the different transfected cells by analyzing the YFP fluorescent intensity of all transfected cells. All cDNA constructs were designed with an YFP tag that is directly coupled to the C-terminal tail of TRPM3 WT, V990M or P1090Q, implying that YFP fluorescence can be used to evaluate total cellular expression levels. Although there was some variation in fluorescence levels between individual cells, no significant differences were observed between WT and the DEE mutants. Altogether, the results suggest that the protein expression level of WT and DEE mutants are within the same range. These data are included in the revised version of the manuscript (Figure 1—figure supplement 1).

When comparing current magnitude between WT and mutants (Figure 1—figure supplement 1), data should be presented as averages of current density, not absolute current, since expression levels surely vary between cells and between mutants. Are the current densities of the maximal currents (i.e., PS+clt-induced currents) comparable for WT and mutant channels? Whereas representative traces were shown, the average current densities under various agonist conditions should be provided.

We thank the reviewers for this comment and have adjusted the figures accordingly. The statistics were performed on current densities, and included in the revised Figure 1—figure supplement 2C-E and G and Figure 4B and F.

2) In Figure 1C, it would be more informative to present the fraction of calcium signal inhibited by primidone instead of -delta Ca.

We appreciated this comment of the reviewers and have tried to re-analyze the results as % inhibition as was suggested. However, the main problem of this type of analysis is the definition of 100% inhibition, since for individual cells it is unknown what their basal calcium concentration would be in the absence of TRPM3 activity. We tentatively defined 100% inhibition of basal activity when primidone reduced the calcium concentration to the mean basal calcium level in non-transfected cells (~30nM), according to:Inh%=100%×[Ca2+]basal−[Ca2+]primidone[Ca2+]basal−mean [Ca2+]basal,NT

However, especially for WT and the P1090Q mutant, this approach often yielded spurious values. In particular, when the basal level before primidone was close to or below the 30 nM level, this approach yielded either negative values or values up to 4000% block (see Author response image 1, top plot). On average, the levels of inhibition calculated in this way were similar for the three channels, at around 70-80 % block (see Author response image 1, bottom). However, considering the limitations described above, we prefer not to include these data in the manuscript.

Instead, we have adapted the graph in Figure 1B, now showing the absolute calcium values under basal conditions and in the presence of the TRPM3 inhibitor primidone (100 µM). In addition, the individual data points of the basal calcium levels before and after primidone are now included in the new Figure 1—figure supplement 1.

3) Regarding the WT:mutant mixing experiments, it’s not clear how these data can be interpreted. The channel composition in humans is unknown; thus assuming that a heterozygous condition implies heteromeric channels is unsupported. The data presented in Figure 4 only indicate that the effects of the mutations can be conferred by a limited number of subunits, if the mutant can form tetramers with WT subunits (in 1:1 ratio), which is also not demonstrated and cannot be assumed a priori. If these data remain in the paper, these possibilities and limitations of the interpretation put forward should be clearly stated and discussed.

We agree that our results neither prove nor disprove the formation of heteromultimers between wild type and mutant subunits. Nevertheless, there are numerous examples in the literature of TRP (e.g. Hoenderop et al. EMBO J 2003) and other tetrameric channels (e.g. MacKinnon Nature, 1991) showing that mixing WT subunits with subunits carrying single point mutations generally leads to the random formation of heteromultimers, following a binomial distribution. Considering that the DEE mutations are point mutations, and not located in domains that are likely to be involved in tetramerisation, we believe it is fair to assume that also here there will be formation of heterotetramers. Currently, there is no knowledge available regarding the exact subunit composition of TRPM3 channels in the cells of human patients. Nevertheless, most dominantly inherited channelopathies caused by point mutations in TRP or other tetrameric channels are attributed to the random formation of heteromultimeric channels with variable subunit composition and altered functionality. In the revised manuscript, we have briefly discussed these possibilities and the limitations of the interpretation.